# Right-to-Left Shunt in Divers with Neurological Decompression Sickness: A Systematic Review and Meta-Analysis

**DOI:** 10.3390/healthcare11101407

**Published:** 2023-05-12

**Authors:** Spyros Peppas, Leonidas Palaiodimos, Sanjana Nagraj, Damianos G. Kokkinidis, Nidhish Tiwari, Amrin Kharawala, Mohammad K. Mojadidi, Sanauallah Mojaddedi, George Ntaios, Robert T. Faillace, Jonathan M. Tobis

**Affiliations:** 1Department of Internal Medicine, MedStar Washington Hospital Center, Washington, DC 20010, USA; 2Department of Medicine, Jacobi Medical Center, Bronx, NY 10461, USA; 3Albert Einstein College of Medicine, Bronx, NY 10461, USA; 4Section of Cardiovascular Medicine, Yale University/Yale New Haven Hospital, New Haven, CT 06520, USA; 5Division of Cardiology, Department of Medicine, Virginia Commonwealth University, Richmond, VA 23284, USA; 6Department of Internal Medicine, Faculty of Medicine, School of Health Sciences, University of Thessaly, 382 21 Larissa, Greece; 7Division of Cardiology, Department of Medicine, University of California, Los Angeles, Los Angeles, CA 90095, USA

**Keywords:** right-to-left shunt, diving, neurological decompression sickness, silent brain lesions

## Abstract

Objective: The aim of this study was to assess the association between the presence of a right-to-left shunt (RLS) and neurological decompression sickness (NDCS) and asymptomatic brain lesions among otherwise healthy divers. Background: Next to drowning, NDCS is the most severe phenotype of diving-related disease and may cause permanent damage to the brain and spinal cord. Several observational reports have described the presence of an RLS as a significant risk factor for neurological complications in divers, ranging from asymptomatic brain lesions to NDCS. Methods: We systematically reviewed the MEDLINE, Embase, and CENTRAL databases from inception until November 2021. A random-effects model was used to compute odds ratios. Results: Nine observational studies consisting of 1830 divers (neurological DCS: 954; healthy divers: 876) were included. RLS was significantly more prevalent in divers with NDCS compared to those without (62.6% vs. 27.3%; odds ratio (OR): 3.83; 95% CI: 2.79–5.27). Regarding RLS size, high-grade RLS was more prevalent in the NDCS group than the no NDCS group (57.8% versus 18.4%; OR: 4.98; 95% CI: 2.86–8.67). Further subgroup analysis revealed a stronger association with the inner ear (OR: 12.13; 95% CI: 8.10–18.17) compared to cerebral (OR: 4.96; 95% CI: 2.43–10.12) and spinal cord (OR: 2.47; 95% CI: 2.74–7.42) DCS. RLS was more prevalent in divers with asymptomatic ischemic brain lesions than those without any lesions (46.0% vs. 38.0%); however, this was not statistically significant (OR: 1.53; 95% CI: 0.80–2.91). Conclusions: RLS, particularly high-grade RLS, is associated with greater risk of NDCS. No statistically significant association between RLS and asymptomatic brain lesions was found.

## 1. Introduction

Decompression sickness (DCS) is defined as the involvement of one or more organ systems caused by gas bubbles released from a previously dissolved state due to a rapid change from a high-pressure environment to a lower ambient pressure [1,2,3]. This is most commonly reported in the context of a rapid ascent after diving but can occur in astronauts during an extravehicular activity in case of sudden depressurization of their suit or while flying in unpressurized aircraft [1]. Symptoms occur due to the supersaturation of tissues with dissolved gas that overwhelms pulmonary filtration capacity or when bubbles enter arterial circulation either through a right-to-left shunt (RLS) or secondary to pulmonary barotrauma [1,4]. DCS is manifested in two major forms; the milder forms include musculoskeletal, cutaneous, and lymphatic DCS, and more severe forms can affect both the cardiopulmonary and nervous systems [5]. Gas and air emboli in coronary circulation can lead to lethal dysrhythmias, myocardial infarction, or cardiac arrest, while nitrogen bubbles in the central nervous system can cause damage via mechanical disruption, vascular obstruction, and promotion of inflammatory pathways and coagulopathy [6,7,8].

The association of patent foramen ovale (PFO) with stroke due to paradoxical embolism of venous blood clots is well-established. The risk of paradoxical embolism and stroke increases with large-sized PFOs, large RLS, and atrial septal aneurysms [9,10]. Neurological decompression sickness (NDCaS) represents a more debilitating presentation of DCS and primarily involves the brain and spinal cord [1,11,12]. The estimated incidence of DCS in recreational, commercial, instruction-led, and/or military divers is 0.9–35.3 per 10,000 person-dives, while more experienced scientific divers have a lower estimated incidence of 0.324 per 10,000 [13]. 

Over the last decade, advances in the prevention, diagnosis, and treatment of NDCS have been stagnant [13]. Several factors can explain the current state. These include the low incidence of NDCS in the general population and the relatively small percentage of the general population that engages in activities that predispose to DCS [13]. On the contrary, PFO is common in the general population, with an estimated prevalence of 25–30% [14,15]. Although the absence of PFO or RLS does not entirely preclude NDCS, its presence in divers, especially when large, has been associated with both asymptomatic brain lesions and NDCS [16]. However, there have been contradictory studies regarding the association between PFO or RLS and brain lesions in asymptomatic divers [17,18,19]. Since the act of diving by itself, even without an RLS, has been associated with asymptomatic brain lesions, the additional risk of developing NDCS is yet to be determined in the presence of commonly occurring RLS [16]. 

Although NDCS is reported to occur rarely, the resulting morbidity is significant and might warrant updated preventive and management strategies. Studies have shown that the presence of PFO increases the risk of DCS by 2.5- to 5-fold, and it is also associated with more severe phenotypes of DCS that may require prolonged hyperbaric treatment [1]. Additionally, previous reports and studies have postulated that divers with PFO, even in the absence of NDCS, could suffer from subclinical brain embolism due to the passage of venous bubbles to arterial circulation, leading to a decline in brain function and detectable brain lesions on MRI [19]. It has been a decade since the Undersea and Hyperbaric Medical Society guidelines on prevention and treatment of decompression illness were last revised [20,21]. Some observational studies on small populations evaluating the association of RLS with NDCS in amateur and professional divers have been conducted since then. The aim of this systematic review and meta-analysis was to consolidate all available data and present an updated and robust analysis that accurately depicts the association between RLS and NDCS and asymptomatic brain lesions.

## 2. Methods

### 2.1. Literature Search

This meta-analysis was performed according to the PRISMA (Preferred Reporting Items for Systematic reviews and Meta-Analyses) guidelines [22]. The review protocol was registered with PROSPERO (CRD42022293821).

A systematic literature search was conducted through MEDLINE, EMBASE and Cochrane CENTRAL until 30 November 2021 for any studies reporting data on the prevalence of RLS in divers with NDCS or with asymptomatic brain lesions. The reference lists of the potential eligible articles and the relevant secondary research studies were reviewed manually. Two investigators (S.P. and L.P.) independently searched for eligible studies. Disagreement on study eligibility were resolved by a third investigator (S.N.) to reach a consensus. The reference lists of pertinent reviews and observational studies were manually searched for more potential studies. A combination of the following keywords was used to perform our search: “PFO”, “patent foramen ovale”, “interatrial shunt”, “atrial septal defect”, “right-to-left shunt”, “left-to-right shunt”, “decompression illness”, “decompression sickness”, “neurological decompression illness”, “cerebral lesion”, “brain lesion”, and “brain damage”. Information regarding the search strategy for each database is mentioned separately in the search protocol.

Any study that fulfilled all the predefined inclusion criteria was considered eligible for this meta-analysis. The inclusion criteria were the following: (i) studies published up to 30 November 2021, (ii) studies comparing the prevalence of RLS in divers with NDCS and inner ear DCS (IEDCS) to that of a control group composed of divers without a prior history of DCS. (iii) studies comparing the prevalence of RLS in healthy divers with asymptomatic brain lesions to those without, and (iv) studies reporting echocardiographic assessment of RLS and detection of brain lesions with cranial magnetic resonance imaging (MRI). The exclusion criteria were as follows: (i) studies with a non-diving cohort; (ii) case reports or case series with fewer than five patients, experimental studies, and reviews; (iii) studies reporting information exclusively on pulmonary or inner ear barotraumas; and (iv) studies not reporting information regarding echocardiographic evaluation of RLS or MRI evaluation for brain lesions. When duplicate studies were identified, the most recent study was included.

### 2.2. Data Extraction and Outcome Measures

The primary outcome was the RLS prevalence in divers with NDCS compared to a healthy diving population. The coprimary outcome was the RLS prevalence in healthy divers with asymptomatic brain lesions compared to those without lesions. The secondary outcome pertained to the degree of RLS as a predictive factor for neurological and NDCS. A subgroup analysis was conducted for the primary outcome of RLS and ΝDCS in divers with cerebral and spinal forms of the disease, along with divers with IEDCS. For the purpose of this article and greater convenience, divers with IEDCS were included in the NDCS group, although it represents a different clinical manifestation of DCS.

Diagnostic modalities used for RLS detection included one of the following methods: transcranial Doppler ultrasonography (TCD), transesophageal echocardiography (TEE), or transthoracic echocardiography (TTE). RLS testing was considered positive by TCD when at least three hyperintense transient signals (HITS) were recorded in the flow of the middle cerebral artery (MCA) within 15–20 s after injection at rest or within 10 s after the release phase of the Valsalva maneuver. TTE and TEE were considered positive for RLS when signals of contrast medium were clearly visualized in the left atrium after complete opacification of the right atrium. However, there was a lack of consistency among included studies regarding the number of cardiac cycles required for the first bubbles to enter the left atrium. Regarding RLS size determination, the presence of >15 HITS on TCD or >20 microbubbles, a cloud of microbubbles, or passage of contrast at rest on TTE or TEE were suggestive of high-grade RLS. The methods used in each study for the diagnosis of RLS are presented in Table 1 and Table 2. A cerebral lesion was considered abnormal if it was hyperintense on proton density-weighted and T2-weighted images and the FLAIR sequence of MRI. 

### 2.3. Risk of Bias Assessment

Two independent reviewers (S.P. and S.N.) assessed studies’ risk of bias using the Quality in Prognosis Studies (QUIPS) tool [23]. Studies were assessed as having a low, moderate, serious, or critical risk of bias for the following domains: study participation, study attrition, prognostic factor measurement, confounding measurement and account, outcome measurement, analysis, and reporting.

### 2.4. Statistical Analysis

We estimated the odds ratios (ORs) and their respective 95% confidence intervals (CIs) for all the individual studies. We performed a meta-analysis using the random-effects model according to the method of DerSimonian and Laird [24]. Heterogeneity among trials for each outcome was assessed with the I^2^ test [25]. In cases in which I^2^ was >75%, the variation across studies was attributed to heterogeneity rather than chance. A forest plot for each outcome was used to graphically display the pooled estimates. Subgroup analysis was performed based on the type of NDCS. The statistical significance level was set at 0.05 with CI calculated at the 95% level. Stata 17.0 (Stata Corp., College Station, TX, USA) was used for statistical analysis.

## 3. Results

The literature search yielded 575 potentially eligible studies after removing duplicates. After screening titles and abstracts, 27 articles were retrieved for full-text evaluation. As shown in the PRISMA flow diagram, fifteen observational studies met the prespecified inclusion criteria. (Figure 1) [12,17,18,19,26,27,28,29,30,31,32,33,34,35,36,37]. One study reported both primary outcomes and was included in both analyses [35]. The study by Germonpre et al. (2021) was not included in our primary analysis because it provided information about the overall incidence of DCS and not NDCS separately [31]. Overall, ten studies were included in the analysis regarding the association between RLS and NDCS [12,26,27,28,32,33,34,35,36,37], whereas six studies were included in the analysis on RLS prevalence in healthy divers with asymptomatic brain lesions [17,18,19,29,30,35]. All included observational studies were assessed as having at least a moderate risk of bias according to the QUIPS tool (Appendix A).

### 3.1. RLS and Neurological DCS

Nine studies reported results on the association between RLS and NDCS [12,26,27,32,33,34,35,36,37]. The total number of patients included in the final dataset was 1902, 954 of whom had a history of NDCS and 876 divers with no history of NDCS. The mean age was 46 years, and the male population accounted for more than 61% of the sample. The most common predisposing factors for NDCS were provocative diving behavior (i.e., rapid ascent, table limit violations, missed decompression stops, and repetitive dives) and concurrent lung disease (Table 1). The overall frequency of RLS in divers with NDCS was 62.6% (597/954), compared with 27.3% (239/876) in divers without NDCS. In this meta-analysis of 10 studies, divers with RLS were at higher risk of NDCS compared to divers without RLS (OR: 3.83; 95% CI: 2.79–5.27; *p* < 0.001; I^2^ = 41.4%; Figure 2).

### 3.2. RLS Size and Neurological DCS

Eight studies reported information on the degree of RLS in study participants. The prevalence of large RLS in divers with NDCS was 51.4% (476/926), whereas in the control group, the prevalence was 16.0% (108/674). Overall, the risk of NDCS was higher in those with high-grade RLS (OR: 4.98; 95% CI: 2.86–8.67; *p* < 0.001 I^2^ = 67.8%; Figure 3) compared to RLS of any grade.

### 3.3. Subgroup Analysis

Subgroup analysis was conducted for studies providing data on the frequency of RLS based on NDCS subtype. Specifically, spinal (N = 6 studies) and cerebral (N = 4 studies) forms, along with IEDCS (N = 3 studies), were identified from the available studies [12,26,33,34,36]. The prevalence rates of RLS in divers with spinal and cerebral forms and IEDCS were 49.5% (189/372), 65.2% (126/193), and 83.6% (204/244), respectively. RLS was associated with a higher risk of IEDCS (OR: 12.13; 95% CI: 8.10–18.17; I^2^ = 0%), compared with cerebral (OR: 4.96; 95% CI: 2.43–10.12; I^2^ = 66.1%) and spinal (OR: 2.47; 95% CI: 1.50–4.07; I^2^ = 47.2%; Figure 4) forms of NDCS.

### 3.4. RLS and Asymptomatic Brain Lesions

Six observational studies were included in the analysis regarding the association between RLS and the occurrence of asymptomatic brain lesions [17,18,19,29,30,35]. The total number of patients included in the analysis was 271. The mean age was 35.4 years, and 88.6% of patients were males. TCD was the primary modality for the assessment of RLS. The baseline characteristics of this population are summarized in Table 2. A total of 69 divers (33.2%) were found to have asymptomatic brain lesions on MRI. Regarding RLS status, 42.0% (29/69) of divers with asymptomatic brain lesions were found to have RLS; in the group without asymptomatic brain lesions, an RLS was present in 38.0% (79/208) of divers. In the meta-analysis of six studies, we did not find an association between RLS and asymptomatic brain lesions in otherwise healthy divers (OR: 1.53; 95% CI: 0.85–2.91; *p* = 0.201; I^2^ = 0%; Figure 5).

### 3.5. Summary of Observational Data

Data derived from studies not included in the final analysis regarding the prevalence of RLS in divers with NDCS are presented in Table 3 [11,16,39,40,41,42,43,44,45,46,47]. 

A retrospective analysis of 209 divers revealed an RLS prevalence of 66.4% in divers with a first episode of cerebral DCS, which reached 100% in divers with ≥2 DCS events. 

A case–control study conducted by Schwerzmann et al. compared the prevalence of NDCS (spinal cord or cerebral form) and asymptomatic brain lesions in divers and non-diving populations with regard to the presence of a PFO. Among the diving cohort, neurological DCS occurred in 4 of 13 divers with and 4 of 39 divers without PFO (OR: 3.0; 95% CI: 1.4–7.2, *p* = 0.03). Similarly, asymptomatic brain lesions were significantly more prevalent in divers with a PFO (1.23 ± 2.0 vs. 0.64 ± 1.22 brain lesions per person, *p* < 0.001) [16].

Overall, five studies reported a significantly high prevalence of hemodynamically relevant RLS in divers with IEDCS, ranging from 73% to 100%, which implies that a selective vulnerability of the inner-ear structure may exist due to the supersaturation of the labyrinth with inert gas longer than in other tissues [11,39,40,41,42]. 

## 4. Discussion

In this systematic review and meta-analysis, we examined the association between (i) RLS and NDCS and (ii) RLS and asymptomatic brain lesions in divers without a history of DCS. The results show that the presence of RLS significantly increases the risk of NDCS, an effect that can be augmented by high-grade RLS. Subgroup analysis demonstrated a higher prevalence of RLS in divers with IEDCS compared with cerebral and spinal forms of the disease. Although there was a trend towards developing asymptomatic brain lesions in healthy divers with RLS, the result did not reach statistical significance.

### 4.1. RLS and Neurological DCS

In the current meta-analysis, we evaluated the association between RLS and NDCS. In addition, a subgroup analysis based on the NDCS type was conducted for the first time and detected a significantly higher prevalence of RLS in divers with IEDCS.

The primary findings of our study are consistent with previous research showing an association between RLS and NDCS. A meta-analysis of five case–control studies compared the prevalence of RLS in divers with NDCS and healthy divers [48]. Lairez et al. reported that divers with RLS were at higher risk of NDCS, with a combined odds ratio of 4.23, an effect that was greater in those with high-grade RLS, with the odds ratio increasing to 6.49 [48]. A similar conclusion but for all DCS cases was reached in the first prospective study conducted to date evaluating the risk of DCS when diving with an RLS [31]. Specifically, among 148 divers included in the final cohort, 28 (18.9%) had a positive carotid Doppler test indicating the presence of an RLS [31]. DCS occurred in a total of 8.3% of RLS-negative divers compared to 28.6% of RLS-positive divers, yielding a total DCS incidence of 1.95 versus 5.16 per 10,000 dives, respectively (RR: 2.65; 95% CI: 1.05–6.72) [31]. Based on the symptoms of DCS, RLS was most commonly associated with vestibulocochlear DCS compared to the spinal type, a finding that aligns with the results of our study and prior literature [12,26,31,34]. 

The association between RLS and NDCS could also be hypothesized by studies demonstrating a significant decrease in related events in divers undergoing transcatheter closure of an RLS [49,50,51,52,53]. In a recent systematic review and meta-analysis of four observational studies including 309 divers (PFO closure group: 141 versus no closure group: 168), PFO closure was associated with a significantly lower incidence of recurrent DCS (PFO closure: 2.84% versus no closure: 11.3%; RR: 0.29; 95% CI: 0.10 to 0.89) [52]. Similarly, a prospective non-randomized study included 29 divers without PFO, 30 divers with PFO but without closure, and 25 divers with PFO who underwent a closure procedure. The study reported significantly lower incidence rates of NDCS and lower numbers of ischemic brain lesions in the no-PFO and PFO closure groups compared to the PFO nonclosure group (0 and 0.5 vs. 35.8 events, 16, and 6 vs. 104 lesions per 10,000 dives, respectively) [49]. 

Although the presence of an RLS has been well-described as a risk factor for NDCS, not all divers with RLS develop the disease; conversely, not all NDCS events are attributed to an RLS [48]. Several diving-related factors could increase the risk of NDCS in the presence of a RLS, such as immersion in cold water, rapid ascent, coexistent lung disease, dives greater than 50 m in depth, or performing the Valsalva maneuver, all of which increase venous return, raising the right atrial pressure and allowing the entrance of venous bubbles into the arterial circulation through an RLS [54,55]. In our study, a significant proportion of included divers exhibited provocative diving behavior, which could increase the likelihood of an NDCS event in divers with RLS. Thus, the presence of an RLS may not represent the main driving force for the development of NDCS, and other diving-related factors (e.g., dives greater than 50 m, immersion in cold water, comorbid lung disease, or provocative diving behavior) should be investigated first.

In addition, regarding RLS-related factors, high-grade RLS was associated with a greater risk of NDCS in our study compared to the mere presence of an RLS. This finding confirms that the impact of small shunts is likely clinically insignificant, as a massive passage of venous bubbles to the arterial circulation through a high-grade RLS probably causes the most severe phenotypes of the disease [26,27,32,36,56]. Wilmshurst et al. evaluated the association between RLS size and shunt-related DCS and reported a greater median size of RLS in divers with a history of DCS compared to the general population (10 mm vs. 5 mm) [57]. Additionally, 50.5% of divers with shunt-related DCS had a PFO diameter greater than 10 mm, while PFO was greater than 10 mm in only 1.3% of the general population [57]. However, PFO diameters in the two groups were measured with different techniques, which may account for this significant difference [57].

Our subgroup analysis demonstrated that divers with an RLS have a greater likelihood of IEDCS compared to cerebral and spinal forms of the disease, which aligns with the results reported in previous retrospective studies and case series [11,34,40,41,42]. A recent case–control study of 639 divers with a history of DCS and 259 healthy control divers reported that 85% of divers with IEDCS were found to have an RLS compared to 32% of non-NDCS divers (OR: 11.8; 95% CI: 7.4–19). The association was weaker with spinal (OR: 2.1; 95% CI: 1.4–3.1) and cerebral (OR: 5.3; 95% CI: 3.2–8.9) forms of the disease [34]. A plausible explanation for the selective vulnerability of the inner-ear structure involves the intravascular bubbles shunted from the venous-to-arterial circulation combined with the prolonged inner-ear inert gas supersaturation compared to other tissues [58]. However, given the overlapping clinical presentation of inner-ear DCS and posterior circulation ischemia from gas emboli, a brain MRI is essential in distinguishing these clinical entities. However, studies evaluating IEDCS cases did not report the usage of MRI for this distinction, and cases of IEDCS may represent cases of cerebellar air embolization. Similarly, it can be challenging to distinguish between IEDCS and inner-ear barotrauma, as both conditions present with similar cochlear and vestibular symptoms. Therefore, clinicians should rely on a detailed history and the clinical features of the presenting illness [59].

### 4.2. RLS and Asymptomatic Brain Lesions

To the best of our knowledge, this is the first analysis conducted to date evaluating the association between the presence of RLS and the occurrence of asymptomatic brain lesions among healthy divers without a prior history of DCS. Previous controlled studies of military and recreational divers support the hypothesis that diving itself increases the risk of more detectable MRI signal abnormalities compared to the healthy, non-diving population [16,29,60], while other reports failed to detect differences between these two populations [61,62,63]. A meta-analysis that compared the prevalence of white matter hyperintensities (WMH) between healthy divers and the non-diving population reported a significantly higher risk of WMH among divers (OR: 2.654; 95% CI: 1.718–4.102), implying that repeated hyperbaric exposure increases the likelihood for the development of subclinical brain lesions [64]. However, no meta-analysis has been conducted evaluating the role of RLS in the development of asymptomatic brain lesions among healthy divers. 

A study of 32 asymptomatic military divers and 32 non-diving healthy controls demonstrated a significantly higher prevalence and number of MRI signal abnormalities in divers compared to controls [29]. In a separate group of divers, the presence of an RLS was associated with a significant difference in the prevalence of focal white matter changes (60% vs. 29.4%; OR: 3.6; 95% CI: 0.8–16), which did not reach statistical significance due to the small number of participants. However, considering the size of the RLS, asymptomatic brain lesions on MRI were found in a higher proportion of divers with high-grade RLS compared to those with a small or no RLS (75% vs. 25%; OR: 9; 95% CI: 1.7–47) [29]. Similarly, in a prospective study of 87 sports divers, PFO-mediated RLS was associated with a higher prevalence of multiple MRI brain lesions (12% vs. 0%), an effect that was greater in association with high-grade RLS (23% vs. 0%) [30]. These findings support the hypothesis that the presence of a clinically significant RLS could induce the formation of multiple subclinical white matter changes in divers, although causality cannot be established yet. However, other studies failed to observe an association between the presence of an RLS and asymptomatic brain lesions [19]. Balestra et al. found no difference in cerebral white matter lesions among divers with or without PFO, and their prevalence was much lower in both groups compared to that reported in other studies [19]. 

## 5. Strengths and Limitations

This is the largest systematic review and meta-analysis conducted to date evaluating the association between (i) RLS and NDCS and (ii) RLS and asymptomatic brain lesions in healthy divers. The main strengths of our study are the strict methodology, robust analysis, and the relatively large number of included studies and overall patient sample, considering the paucity of available studies on the diving population. In addition, for the first time, a subgroup analysis was conducted based on the type of NDCS, which confirmed the stronger association between RLS and IEDCS.

Our study is subject to several limitations. First, there was heterogeneity in the detection method used for RLS diagnosis among individual studies, which may account for differences in RLS prevalence across studies. Second, because the diagnosis of neurological DCS is based on the objective description of patient events, the number of true disease cases may be underestimated in our study. This is more evident regarding inner-ear DCS cases, which may represent cases of cerebellar gas embolization, as these two distinct conditions can present with overlapping clinical features. The rapid onset of inner-ear symptoms with normal videonystagmography should raise suspicion for a central lesion, which requires an emergency brain MRI to exclude the possibility of a cerebellar infarction [65]. However, MRI was not used in the studies included in our subgroup analysis. Third, regarding the association of RLS and asymptomatic brain lesions in healthy divers, the small sample of the diving population may account for the absence of a statistically significant difference in RLS prevalence between divers with asymptomatic brain lesions and those without, although there was a trend towards a higher RLS prevalence in the former group. Fourth, diving settings (e.g., recreational or amateur vs. professional or military), along with diving experience, represent factors that can influence the association between PFO and neurological decompression sickness. As only a minority of studies reported the diving settings, diving experience, and how they affected the occurrence of NDCS, we were unable to adjust for these factors. Lastly, the observational nature of the included studies makes them prone to selection and confounding bias. 

## 6. Conclusions

This systematic review and meta-analysis demonstrates a higher risk of NDCS in divers with RLS, which is more pronounced in association with the presence of a high-grade RLS. The subgroup analysis based on neurological symptoms showed a stronger association with IEDCS. Although there was a trend towards developing asymptomatic brain lesions in healthy divers, the effect did not reach statistical significance.

Despite these limitations, our findings suggest that RLS, particularly high-grade RLS, represent a significant risk factor for NDCS. However, further prospective studies with adequate power and adjustment for confounding variables are warranted to evaluate RLS as the major driving force for severe DCS phenotypes. Since the absolute incidence of NDCS is low, we do not recommend routine RLS screening for all divers. Divers with a history of a DCS episode are recommended to undergo RLS screening with TTE, a highly sensitive, non-invasive alternative to TEE that should be utilized as the test of choice for detection of RLS in future studies [66]. Following a diagnosis of PFO-mediated RLS, a diver, in consultation with a diving physician, could either adopt conservative diving strategies or undergo transcatheter PFO closure to prevent DCS events in the future. In addition, for future studies evaluating the association between RLS and IEDCS, we advocate for the utilization of brain MRI in divers presenting with inner-ear-type symptoms to exclude the possibility of gas emboli in posterior circulation. 

## Figures and Tables

**Figure 1 healthcare-11-01407-f001:**
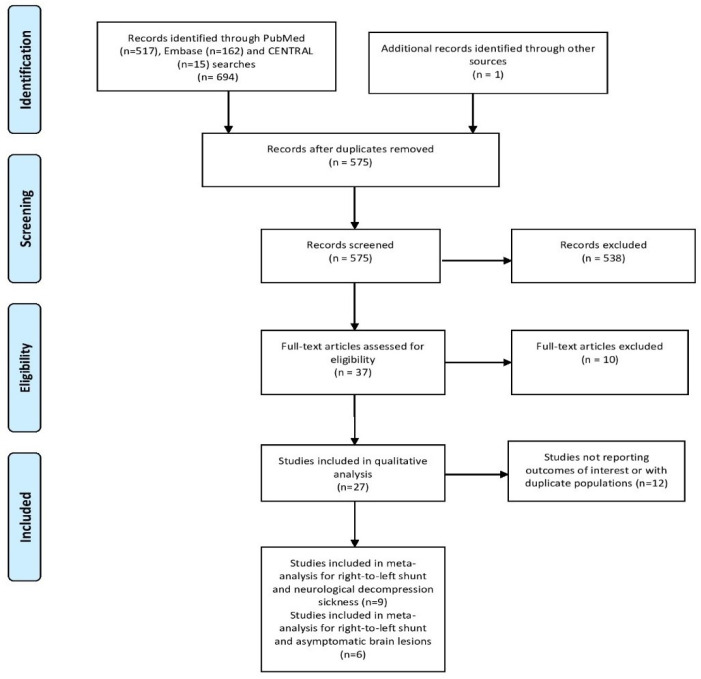
Flow chart of the meta-analysis [38].

**Figure 2 healthcare-11-01407-f002:**
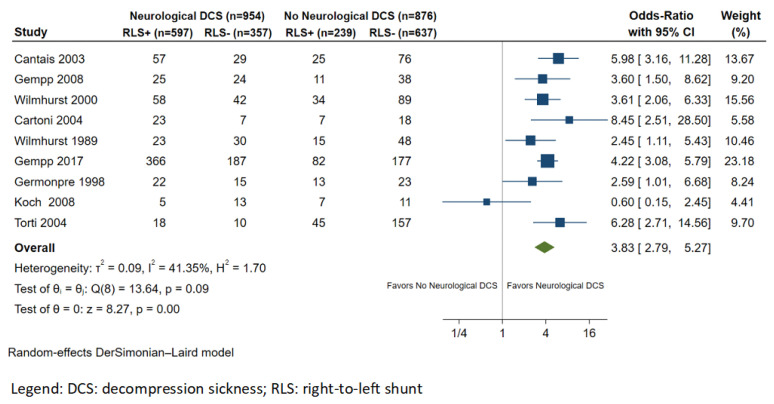
Forest plot of the meta-analysis comparing right-to-left shunt prevalence between divers with and without neurological decompression sickness.

**Figure 3 healthcare-11-01407-f003:**
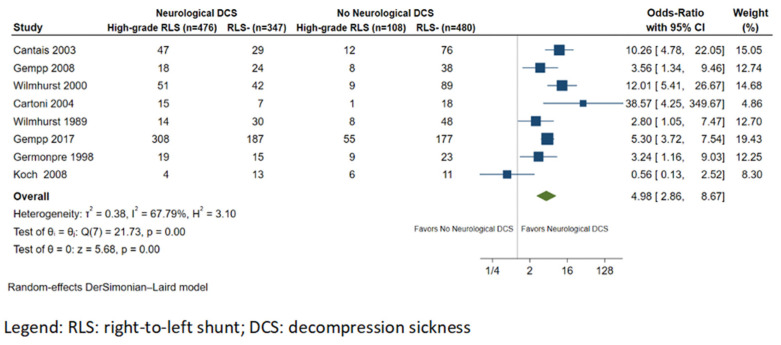
Forest plot of the meta-analysis comparing the prevalence of high-grade right-to-left shunts between divers with and without neurological decompression sickness.

**Figure 4 healthcare-11-01407-f004:**
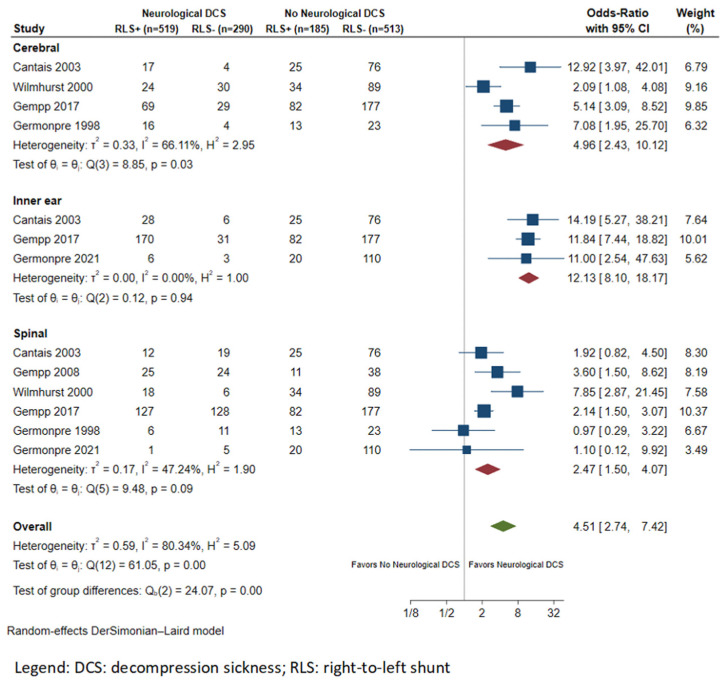
Forest plot of the meta-analysis comparing right-to-left shunt prevalence between divers with and without neurological decompression sickness according to disease subtype.

**Figure 5 healthcare-11-01407-f005:**
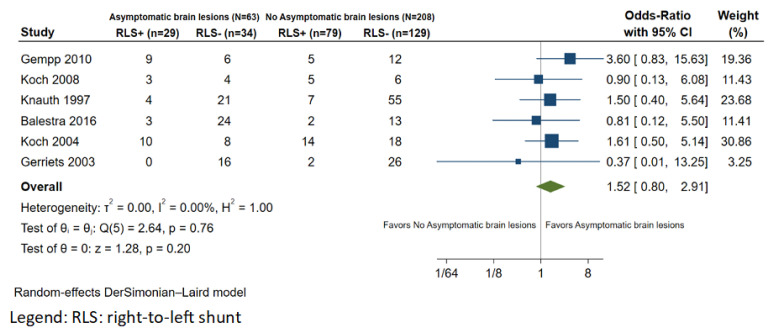
Forest plot of the meta-analysis comparing right-to-left shunt prevalence between divers with and without asymptomatic brain lesions.

**Table 1 healthcare-11-01407-t001:** Baseline characteristics of the diving population included in our analysis regarding the association between right-to-left shunts and neurological decompression sickness.

Study (Author, Year)	NDCS Group, N	No NDCS Group, N	Professional Divers	Age, Years	Males, %	N of Dives	Depth of Diving (m)	Diving Experience, Years	NDCS Type	Predisposing Factors for NDCS (N)
Case–control studies										
Cantais, 2003	86 (101 DCS patients)	101	Ν/A	35 ± 10.3 (all DCS patients) ^a^ 33 ± 9.3 (no NDCS) ^a^	84.1 (all DCS patients) 76.2 (no NDCS)	N/A	>30 m (n = 63) (NDCS group)	Ν/A	Cerebral (21) Spinal (31) Inner ear (34)	Table limits violations (46)
Gempp, 2008	49	49	N/A	46 ± 12 (NDCS) ^a^ 41 ± 7 (no NDCS) ^a^	77.6 (NDCS) 84.7 (no NDCS)	N/A	40 ± 11 (NDCS group)^a^	N/A	Spinal (49)	No
Wilmshurst, 2000	100	123	9 (NDCS group)	N/A	66 (NDCS)	N/A	N/A	N/A	Cerebral (54) Spinal (24) Combined (14) Indeterminate (8)	Lung disease, rapid ascent, missed deco stops (30) (PFO−) No (PFO+)
Wilmshurst, 1989	53 (61 DCS patients)	63	2 (DCS patients) 0 (no NDCS)	Ν/A	77.4 (NDCS) 80.3 (no NDCS)	Ν/A	N/A	N/A	Ν/A	Rapid ascent, missed deco stop, dive > 50 m, repeat diving (31 DCS episodes)
Gempp, 2017	553 (634 DCI patients)	259	N/A	43.6 ± 11.3 (all DCS patients) ^a^ 34.6 ± 9.0 (no NDCS) ^a^	N/A	N/A	N/A	Ν/A	Cerebral (97) Spinal (255) Inner ear (201)	Ν/A
Germonpre, 1998	37	36	N/A	37.5 ± 9(NDCS) ^a^	N/A	327 ± 282 (cerebral) ^a^ 481 ± 465 (spinal) ^a^	35 ± 11 (cerebral) 41 ± 8 (spinal)^a^	8 ± 6 (cerebral) ^a^ 12 ± 10 (spinal) ^a^	Cerebral (20) Spinal (17)	Fault during diving: (8, cerebral), (3, spinal)
Koch, 2008	18	18	N/A	37 ± 10.2 (NDCS) ^a^ 40.3 ± 12.7 (no NDCS) ^a^	83.3 (NDCS) 83.3 (no NDCS)	888 (7–6000) (NDCS) ^b^ 870 (110–5500) (no NDCS) ^b^	N/A	N/A	N/A	No
Cross sectional/cohort studies										
Cartoni, 2004	30 (41 DCS patients)	25	All	35 ± 8 (DCS patients) ^a^	90.9 (all)	N/A	42 ± 11 (RLS+) ^a^ 31 ± 11 (RLS-) ^a^	N/A	N/A	Repetitive dives (10) Missed deco stops/rapid ascent (21)
Torti, 2004	28	202	N/A	39 ± 8 (all) ^a^	80 (all)	650 (250–1200) (RLS+) ^c^ 400 (214–800) (RLS−) ^c^	29 ± 9 (RLS+) ^a^ 28 ± 9 (RLS−) ^a^	11 ± 8(RLS+) 9 ± 7 (RLS−) ^a^	N/A	No
Liou, 2015	32	43 *	N/A	39 ± 13 (all) ^a^	61 (all)	100 ± 178 (RLS+) ^a^ 65 ± 202 (RLS−) ^a^	29 ± 10 (RLS+) ^a^ 32 ± 28 (RLS−) ^a^	N/A	N/A	N/A
Germonpre, 2021	18 (all DCS) **	130	N/A	38.3	68 (all)	441.0 ± 751.2 (RLS+) ^a^ 524.7 ± 843.56 (RLS−) ^a^	N/A	N/A	Inner ear (9) Spinal (6)	Most cases in PFO+ group refer to provocative dives
**Study**	**Diagnostic Modality**	**Criteria for the Diagnosis of RLS**			**Criteria for the Diagnosis of High-Grade RLS**		**RLS in NDCS Group, N (%)**	**RLS in No NDCS Group, N (%)**	**Large RLS in NDCS Group, N (%)**	**Large RLS in no NDCS Group, N (%)**
Case–control studies										
Wilmshurst, 1989	TTE	Passage of microbubbles in the L atrium within two cardiac cycles after complete opacification of the R atrium			>20 microbubbles		23 (43.4)	15 (23.8)	14 (26.4)	8 (12.7)
Germonpre, 1998	TEE	Passage of microbubbles in the L atrium within three heart cycles after complete opacification of the R atrium			≥20 microbubbles at rest or after Valsalva strain		22 (59.5)	13 (36.1)	19 (51.3)	9 (25)
Wilmshurst, 2000	TTE	Passage of microbubbles in the L atrium after the first injection at rest or up to five injections performed with the Valsalva maneuver			>20 microbubbles		58 (58)	34 (27.6)	51 (51)	9 (7.3)
Cantais, 2003	TCD	>5 HITS 5–15 s after injection			>20 HITS within 20 s		57 (66.3)	25 (24.8)	47 (54.7)	12 (11.9)
Cartoni, 2004	TEE/TTE	≥3 microbubbles within three cardiac cycles			Shunts occurring at rest		23 (76.7)	7 (28)	15 (50)	1 (4)
Gempp, 2008	TCD	>5 HITS within 20 s after injection or <10 s after release phase			>20 HITS		25 (51)	11 (22)	18 (37)	8 (16)
Gempp, 2017	TCD	>5 HITS within 15 s after normal breathing or 10 s after the end of provocative maneuver			>20 HITS		366 (66.2)	82 (31.7)	308 (55.7)	55 (21.2)
Cross-sectional studies										
Torti, 2004	TEE	Passage of bubbles from the R to L atrium within four cardiac cycles			Visualization of a cloud of bubbles		18 (64.3)	45 (22.3)	N/A	N/A
Koch, 2008	TCD, TTE	Signals of contrast visualized in the L atrium after complete opacification of the R atrium OR > 5 HITS within 10 s after Valsalva release			>20 HITS or spontaneous shunting		5 (27.8)	7 (38.9)	4 (22.2)	6 (33.3)
Liou, 2015	TTE	Passage of bubbles from the R to L atrium within four cardiac cycles			N/A		23 (71.9)	16 (37.2)	N/A	N/A
Germonpre, 2021	Carotid Doppler	Visualization of microbubbles after up to three injections with straining maneuvers			N/A		6 (Inner ear) (66.6) 1 (Spinal) (20)	20 (15.4)	N/A	N/A

^a^ mean ± SD; ^b^ mean (range); ^c^ median (IQR); * patients in control group had minor DCS; ** information about overall neurological DCS cases was not available; RLS: right-to-left shunt; NDCS: neurological decompression sickness; N: number; m: meters; DCS: decompression sickness; deco: decompression; N/A: not available; TCD: transcranial Doppler; TEE: transesophageal echocardiography; TTE: transthoracic echocardiography; HITS: high-intensity transient signals; R: right; L: left.

**Table 2 healthcare-11-01407-t002:** Baseline characteristics of diving population included in our analysis regarding the association between right-to-left shunts and asymptomatic brain lesions.

Study (Author, Year)	Design	Divers, N	ABLs Group, N (%)	No ABLs Group, N (%)	Professional Divers, N	Age, Years	Males, %	Diving Depth (m)	N of Dives
Knauth, 1997	Cross sectional	87	11 (12.6)	76 (87.4)	All amateur	35.7 ± 8.9 ^b^	77.0	N/A	565.3 ± 509.1 ^b^
Gerriets, 2003	Cross sectional	42	1 (2.4)	41 (97.6)	All amateur	35.7 ± 7.9 ^b^ (RLS+) 32.2 ± 7.5 ^b^ (RLS-)	97.6	27.7 (15.5–39.8) ^c^	305 (20–4970) (RLS+) ^a^ 295 (21–2000) (RLS-) ^a^
Koch, 2004	Cross sectional	50	24 (48)	26 (52)	Military & Civilian	34.7 ± 10.6 ^b^	94	N/A	500 (21–5500) ^a^
Koch, 2008	Case–control	18	8 (44.4)	10 (55.6)	Military	40.3 ± 12.7 ^b^	83.3	N/A	870 (110–5500) ^a^
Gempp, 2010	Case–control	32	14 (43.7)	18 (56.3)	Military	35 ± 5 ^b^	100	All < 60 m	1659 ± 122 ^b^
Balestra, 2016	Cross sectional	42	5 (11.9)	37 (88.1)	All amateur	36 ± 4.85 ^b^	90.5	57.9% dives < 30 m	620 ± 465 ^b^
**Study (Author, Year)**	**Diagnostic Modalities (RLS/ABLs)**	**Criteria for the Diagnosis of RLS**	**Criteria for the Diagnosis of High-Grade RLS**	**RLS in ABLs Group,** **N (%)**	**RLS in No ABLs Group,** **N (%)**	**Large RLS in ABLs Group,** **N (%)**		**Large RLS in No ABLs Group,** **N (%)**	
Knauth, 1997	TCD/MRI	≥5 HITS after the Valsalva maneuver	≥20 HITS	4/11 (36.4)	21/76 (27.6)	N/A		N/A	
Gerriets, 2003	TCD/MRI	>3 HITS	≥20 HITS	0/1 (0)	16/41 (39)	N/A		N/A	
Koch, 2004	TCD, TTE/MRI	Signals of contrast visualized in the L atrium after complete opacification of the R atrium OR >5 HITS within 10 s after Valsalva release	N/A	10/24 (41.7)	8/26 (30.8)	N/A		N/A	
Koch, 2008	TCD, TTE/MRI	Signals of contrast visualized in the L atrium after complete opacification of the R atrium OR >5 HITS within 10 s after Valsalva release	>20 HITS or spontaneous shunting	3/8 (37.5)	4/10 (40)	N/A		N/A	
Gempp, 2010	TCD/MRI	>5 HITS within 20 s after injection or 10 s after the release phase of the Valsalva maneuver	>15 HITS	9/14 (64.3)	6/18 (33.3)	9/14 (64.3)		3/18 (16.6)	
Balestra, 2016	TEE/MRI	Passage of microbubbles in the L atrium within three heart cycles after complete opacification of R atrium	>20 microbubbles	3/5 (60)	24/37 (64.9)	2/5 (40)		14/37 (37.8)	

^a^ Median (range); ^b^ mean ± SD; ^c^ mean (range); N: number; RLS: right-to-left shunt; ABLs: asymptomatic brain lesions; TCD: transcranial Doppler; TTE: transthoracic echocardiography; TEE: transesophageal echocardiography; MRI: magnetic resonance imaging; N/A: not applicable; m: meters; HITS: high-intensity transient signals; R: right; L: left.

**Table 3 healthcare-11-01407-t003:** Summary of observational data not included in the analysis regarding the association between right-to-left shunts and neurological decompression sickness.

Study (Author, Year)	Design	Ν of Divers	Age, Years (mean ± SD)	Males, N (%)	Predisposing Factors for NDCS (N)	Diagnostic Modality	Divers with NDCS, N	NDCS Type (N)	RLS in Divers with NDCS, N (%)	Large RLS in Divers with NDCS, N (%)
Moon, 1989	Case–control	30	33.5 (12–48)	23 (76.6)	N/A	TTE	18	N/A	11 (61.1)	N/A
Kerut, 1997	Retrospective cohort	26	29.1 ± 6.4	23 (88.4)	N/A	TEE	15 *	N/A	9 (60)	N/A
Schwerzmann, 2001 **	Retrospective cohort	52	38 ± 10 (RLS group) 35 ± 8 (no RLS group)	10 (77) (RLS group) 30 (77) (no RLS group)	N/A	TEE	4/13 (RLS group) 4/39 (no RLS group)	Spinal or cerebral	N/A	N/A
Klingmann 2003	Case series	9	N/A	N/A	No	TCD	9	IEDCS	9 (100)	9 (100)
Klingmann, 2007	Retrospective cohort	18	43 (25–61)	15 (83.3)	N/A	TCD	18	IEDCS	15 (83.3)	15 (83.3)
Harrah, 2008	Retrospective cohort	113	40 (19–67)	83%	N/A	TTE	48	N/A	6/12 (50)	N/A
Klingmann, 2012	Retrospective cohort	30	43 ± 9 (25–60)	23 (76.6)	Repetitive diving (26)	TCD	30	IEDCS	22 (73)	N/A
Gempp, 2012	Retrospective cohort	115	44 ± 11	99 (86)	Provocative decompression schedule (4) Repetitive dives within 24 h (38)	TCD	115	IEDCS (all)	95 (82.6)	89 (77)
Ignatescu, 2012	Retrospective cohort	33	46 (31–61)	31 (94)	Decompression diving or dehydration (22) Previous DCS episode (6)	TTE	33	IEDCS (all)	24/30 (80)	23 (76.6)
Guenzani, 2016	Case series	9	47.3	9 (100)	Omitted stops (2)	N/A	9	IEDCS (all)	5 (55.5)	N/A
Lafere, 2017	Retrospective cohort	209	40.5 ± 11.2	80.4%	N/A	TEE/TTE	209	Cerebral (all)	167 (80)	133 (63.6)

* Patients with probable or definite NDCS diagnosis; ** data from this cohort of patients were published by Torti et al. (2004) [27]. N: number; RLS: right-to-left shunt; NDCS: neurological decompression sickness; TTE: transthoracic echocardiography; TEE: transesophageal echocardiography; TCD: transcranial Doppler; SD: standard deviation.

## Data Availability

No data were created.

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
