# Peer review of "Right-to-Left Shunt in Divers with Neurological Decompression Sickness: A Systematic Review and Meta-Analysis"

_healthcare, 2023, doi:10.3390/healthcare11101407_

Round 1
Reviewer 1 Report
Dear authors,
Thank you for your submitted manuscript. It is an interesting topic to me.
Overall, it has well-designed methodology, results report, and especially the discussion and conclusion which addressed that right to left shunt (RLS) should not be overemphaszed as a significant risk for neurological complications in decompression sickness (DCS). I strongly agree with this comment addressed. Herein, please consider my comments and suggestions as below:
1. What are the rationales or significance of RLS contibuting to DCS in supporting to do this study, such as its high prevalence or high risk of NDCS occurrence, though some part of them were described in introduction. Kindly the authors expand the content for more details.
2. The authors used the term 'right-to left shunt (RLS)' which refered to abnormal intracardiac blood flow direction, not just only an anatomical defect of cardiac septum. Whther the authors included only the studies in which the study cases had such intracardiac shunt confirmed, and by what investigation methods (e.g. color-doppler ecchocardiogram or else).
3. 'Left to right shunt' was also found as a search term in this study. What(s) is (are) the reason(s)?
4. Whether only studies reporting interatrial septal defect, or either or both inter'artial' and inter'ventricular' septal defect were included in the meta-analyis.
5. As PFO is commonly found in general population, and mostly without clinical significance, whether the authors focused largely on the high risk of DCS occurrence attributed to PFO (line 53-73) (or other septal defect with RLS) existence in this study. Please clarify.
6. Violation of diving safety protocols in recreation or sport divings is the powerful contributor for DSC . The difference settings of diving (recreational, military prupose or making a living) and the experience of the divers significantly effect the DCS occurrence. Are these factors taken into consideration, or adjusted during the analysis?
7. If it is possible to limit the diving setting in this study to only one kind, i.e sport diving in which safety diving protocols are widely applied, whether there still is as high association of RLF with DCS. (This is only my comment and interest).
8. Some minor errors in typesetting and writting styles were observed, I would like to suggest as the followings:
8.1 The use of an abbreviation such as RLS and DCS should be prceded by the full-written term (abbreviation). Please check them throughout.
8.2 Please clearly define and consistently use the terms like 'neurological DCS' whether the authors intended to mean a) only brain and/or spinal cord, and b) the etiology of neurologcal complications limited only to ischemic stroke or else?
8.3 For the term 'healthy' or 'healthy divers' used in many places. I guess the authors intended to mean the study participants who had 'no clinical NDCS'. I would like to suggest the 'NDCS(+)', or 'NDCS(-)' for those who had, or had no NDCS.
Otherwise, please define the terms and use them consistently through out.
8.4 The term' brain lesion'. Did the authors mean only 'ischemic brain lesion' or also others, please clarified or specified.
8.5 Whether 'silent brain lesion' (line 21) was equivalent to 'asymptomatic brain lesion' Please consistently use the term.
8.6 In the Figure 1 Flowshart......, the final number of studies for analysis in the lowerest box is '15', while the authors declared '10' studies were included for final meta-analysis. Please recheck or complete the content of the flowchart if it was missed.
8.7 Page aligment of all tables should be revised in horizontal plane for being easily readable.
8.8 Line 349-350, probably it should be "the true number of disease cases may be 'under'estimated (not overestimated) in our study." -- I think the authors intended to say that the reporting number of disease cases was much lower than a true (or real) one !? Please recheck and cnfirm.
8.9 Line 374 in conclusion, ".....undergo RLS screening with TCD,". I think may be with ecchocardiography (transthoracic method)- not transcranial doppler (TCD), hence please recheck whether it should be 'TTE'.
8.10 Inner ear DCS was attributed to possible vertebro-cerebellar stroke, rather than barotruama against the inner ear as having been known, in the discussion of this study and MRI brain was recommended to confirm the diagnosis in future study. Could the authors add more information and refer to the previous studies that suggested this etiology.
Thank you
Reviewer 2 Report
The authors present a large systematic review and meta-analysis of the association between right-to-left shunts (RLS) in decompression sickness ( DCS) in divers. DCS is not very common in the general population, so the disease is most likely underestimated and under-investigated. The authors conducted a rigorous literature search of the available literature, including studies from 1989-2021. The authors confirm that the presence of a RLS is associated with a higher risk of neurological DCS, and larger shunts even increase this risk. Subgroup analysis revealed a significantly higher prevalance of RLS in divers with inner ear DCS. The main limitation is the lack of brain MRI data in many of the investigated studies. However, the findings are valid and interesting, given the paucity of available studies on this topic. The authors recommend, since the absolute incidence of neurological DSC is low in divers, against routine screening and -closure for RLS, as the diving behaviour can also influence the incidence of DCS.
Further prospective studies with MRI data are required to clarify the open questions.
In summary, it is an interesting systematic review of an underinvestigated topic. The limitations are given by the paucity of available data, but this cannot be influenced by the authors.
Reviewer 3 Report
Compliments to all the Authors.
The argument treated in the paper is not original, nevertheless it is well-defined and providing an interesting update of the current knowledge (expecially as per the visual representation of the findings to date, thanks to well-designed Forest Plots depicting the results of the meta-analysis).
About what stated at line #309-311, there is at least a previous analysis on the association RLS-DCS ("Risk of Neurological Decompression Sickness in the Diver With a Right-to-Left Shunt: Literature Review and Meta-Analysis, by Olivier Lairez et al. in Clin J Sport Med - Volume 19, Number 3, May 2009"; ref.: https://www.uhms.org/images/DCS-and-AGE-Journal-Watch/lairez_umo_neurological_deco.pdf). The results presented in the paper submitted are anyhow significant and interpreted appropriately. On my opinion, as per the method adopted in the bubbles detection, further analysis would probably have to shift to the sole transcranial Doppler (TCD), appearing to be the methodological approach less operator-dependant.
Minor revisions suggested: avoid extra spaces and misjustification as per lines #: 50 and 498; 98-99 and 477.
At lines 170-173 I'm proposing to include in the bibliography either "Hyperbaric oxygen for decompression sickness: 2021 update. Richard E. Moon, and Simon J. Mitchell MD. UHM 2021, VOL. 48 NO. 2" or the following update these Authors (and MH Bennett) recently wrote for the NEJM: "Decompression Sickness and Arterial Gas Embolism. Mitchell SJ, Bennett MH, Moon RE. N Engl J Med. 2022 Mar 31;386(13):1254-1264. doi: 10.1056/NEJMra2116554".
Final personal consideration: in the conclusions it could be an enriching add-on for the reader any Authors' revisiting (thanks to the data actually presented) the statement proposed by SPUMS on the topic [Persistent Foramen Ovale (PFO) and Diving, 2015: https://spums.au/images/PositionStatements/2015-06%20SPUMS_Joint%20positionPW.pdf].06%20SPUMS_Joint%20positionPW.pdf].
Reviewer 4 Report
This systematic review addresses an important topic, and the meta-analysis is well performed. The results are clearly presented and the discussion understandably interprets the results.
There are only some minor comments that are listed below:
There is a tendency to avoid the DCS I and DCS II classification, because no classes are to be treated but diseases. Hence, ‘serious’ and ‘mild’ symptoms more intuitively describe the event.
Pls, see also Richard D Vann DOI: 10.3357/ASEM.2471.2009
MEDLINE in the Abstract and later PubMed. Pls, unify.
l. 95 -99: no search term for ‘diving’?
l. 145: pls, reference …. Later DerSimonian. Pls, unify.
l. 157: there two studies by Germonpé. Thus: One study ….
Table 1 and Table 2 and Table 3: is there an order, by year or alphabet?
l. 232: …. examined the association between i) RLS and neurological DCS and ii) RLS and asymptomatic brain lesions in healthy divers.
The above sentence - to this reviewer - suggests that aim i) includes sick divers and aim ii) healthy ones. Could you, pls, improve?
l. 248: Who They? Healthy divers?
p. 12, 2. Para: Maybe speculate on reasons, why even after PFO closures DCS happens. E.g. pulmonary shunts? And what about a warning: PFO closure does warrant DCS absence.
p. 12, 3. Para: my above concern is resolved. Sorry, for any confusion.
l. 296: Maybe add: .. from other previous ….
